# Are Opportunities Being Missed? Burden of HIV, STI and TB, and Unawareness of HIV among African Migrants

**DOI:** 10.3390/ijerph16152710

**Published:** 2019-07-30

**Authors:** Sónia Dias, Ana Gama, Ana Maria Tavares, Vera Reigado, Daniel Simões, Emília Carreiras, Cristina Mora, Andreia Pinto Ferreira

**Affiliations:** 1Escola Nacional de Saúde Pública, Centro de Investigação em Saúde Pública, Universidade NOVA de Lisboa, 1600-560 Lisboa, Portugal; 2Global Health and Tropical Medicine, GHTM, Instituto de Higiene e Medicina Tropical, IHMT, Universidade Nova de Lisboa, UNL, 1349-008 Lisboa, Portugal; 3GAT—Grupo de Ativistas em Tratamentos, 1000-228 Lisboa, Portugal; 4AJPAS—Associação de Intervenção Comunitária, Desenvolvimento Social e de Saúde, 2700-327 Amadora, Portugal; 5Ser + - Associação Portuguesa para a Prevenção e Desafio à Sida, 2750-783 Cascais, Portugal

**Keywords:** sub-Saharan African migrants, HIV, STI, TB, unawareness of HIV, HIV testing

## Abstract

Sub-Saharan African migrants (SSAMs) have been disproportionately affected by infectious disease burden. We aimed to identify correlates of HIV, past sexually transmitted infection (STI) and past Tuberculosis infection (TB), as well as examine HIV seropositivity unawareness and testing history among SSAMs. A venue-based sample of 790 SSAMs completed a cross-sectional biobehavioral survey on sexual practices, HIV testing and self-reported infectious diseases; an HIV rapid test was offered. Overall, 5.4% of participants were HIV-positive and 16.7% reported a past STI. Odds of being HIV positive or having a past STI were higher among participants with low socioeconomic status and who experienced violence from a partner. Increased odds of having a past STI were also found among long-term migrants and those who reported sexual risk behaviors. In total, 4.1% of participants had TB in the past; these were more likely male and HIV positive. Unawareness of HIV-positive status was notably high (35%). Half of the participants had never been tested for HIV before, including over a third of those who had STI or TB in the past. Efforts are needed to reduce missed opportunities for HIV/STIs prevention and uptake of HIV testing among SSAMs through more integrated care, while addressing social determinants of infectious diseases.

## 1. Introduction

Increased rates of migration to and within Europe in recent years have made migrant health a priority for the region [1,2]. In 2017, 78 million international migrants were residing in Europe [3], and 9 million originated from African countries [4]. In a number of European Union/European Economic Area Member States, subgroups of migrant populations including those from African countries are disproportionately affected by infectious diseases such as HIV, other sexually transmitted infections (STIs) and Tuberculosis infection (TB) [1,5,6,7,8]. Due to historical reasons, Portugal has been a host country of many sub-Saharan African migrants (SSAMs) [9], mostly from Portuguese-speaking African countries [10]. In 2017, a third of the HIV cases diagnosed in the country (340/1064) was among foreign-born individuals; SSAMs represented 57.6% of those cases (*n* = 196) [11]. Recent data have also shown that the proportion of TB cases among foreign-born individuals in Portugal has been increasing (19% in 2017 vs. 15.9% in 2014) [8,12], although no information was available on the countries or regions of origin.

The vulnerability of migrant populations to infectious diseases is likely shaped by multilevel factors that increase exposure to risk through their migratory environments [13,14]. The prevalence of infectious diseases such as HIV, other STIs and TB has been associated with the high disease burden in the countries of origin [14,15,16]. Other factors manifest in the host countries and include the poor socioeconomic conditions frequently faced by migrant populations upon arrival and during their stay [11,12,17,18]. The context of socioeconomic disadvantage may also pose barriers in access to health services for information, prevention and care, delaying diagnosis and treatment, and increasing the risk for transmission, morbidity and mortality [19,20,21]. During migration, the separation from partners and the sense of solitude and isolation experienced by many migrants may favor the engagement in risky sexual practices, putting them particularly at increased risk for HIV and STIs [22]. The migration process may also expose individuals to sexual and gender-based violence, often involving unprotected sexual contact [23]. Additionally, having an infectious disease may increase the risk for acquiring other infectious diseases. This is the case with syphilis and gonorrhea which increase the risk of HIV infection [24], and also the case of HIV/AIDS which increase the risk of TB infection, activation or reactivation [25].

Research on infectious diseases and related factors among migrant populations is critical to further understand these populations’ health vulnerabilities and inform prevention strategies. However, migrants are often underrepresented in national statistics and research [26]. The burden of infectious diseases among migrants from sub-Saharan African countries in particular has been understudied. This study aimed to identify correlates of HIV, past STI and past TB among a population of SSAMs living in Lisbon, as well as examine unawareness of HIV seropositivity and testing history in this population.

## 2. Materials and Methods

### 2.1. Study Design and Participants

A cross-sectional bio-behavioral study was conducted with SSAMs in Lisbon. Most recent official data indicate that 18,699 SSAMs with legal resident status were living in Lisbon, which is the region with the highest concentration of immigrants in the country [10,27]. A participatory approach was adopted with the active collaboration of non-governmental organizations (NGOs) with years of community work experience in different phases of the study [28].

A venue-based sampling method [29] was used to recruit participants. Methods and procedures used are described in detail elsewhere [30]. Through formative research in partnership with community members, geographic and social mapping was developed in Lisbon region to systematically identify and list venues commonly frequented by SSAMs (e.g., associations, neighborhoods, events). Recruitment teams of outreach workers from partner NGOs and migrant peers attended the listed venues to approach potential participants and to invite them to participate in the study. The inclusion criteria were being ≥18 years old, being a SSAM, residing in Portugal, and having initiated sexual activity.

### 2.2. Data Collection

In brief, data were collected at recruitment sites, in appropriate spaces providing privacy and comfort to the participants. An anonymous structured questionnaire was applied by interviewers. Interviewers were recruited from community-based organization partners and received specific training on the study purpose, data collection procedures and ethical considerations.

After the questionnaire, a finger-stick whole-blood HIV-rapid test was offered and accepted by all participants, and was performed by qualified technicians of the partner NGOs. Those with a reactive test were referred to healthcare services for confirmatory testing and linkage to care.

Anonymous participation and confidentiality of data were guaranteed. Informed consent was obtained from all participants. Participants received free condoms and leaflets with information on prevention, testing and treatment. The study was approved by the Ethics Committee of Instituto de Higiene e Medicina Tropical, Universidade Nova de Lisboa (09-PN-2014).

### 2.3. Measures

The questionnaire included close-ended questions on sociodemographic characteristics, sexual practices, HIV testing and self-reported infectious diseases. Sociodemographic characteristics included sex, age, educational level, occupation, perceived income and length of stay in Portugal.

Participants who were sexually active in the last 12 months were asked about their sexual practices for this reference period. Sexual practices assessed were number of sexual partners, condom use (“consistent” included the option “always used condom”; “inconsistent” included the options “sometimes” and “never” used condom) and engagement in commercial sex (defined as having received or paid money/goods in exchange for sexual services as a sex worker or as a client). Participants were also asked about experience in their lifetime of physical/emotional violence from a partner and of forced sexual relations.

Participants were asked if they ever tested for HIV and about their current status for HIV infection based on the last HIV testing result. Participants were also asked if they ever had or had not been diagnosed with other sexually transmitted infections (than HIV) and TB in their lifetime. Unawareness of HIV infection was defined as having a reactive HIV rapid test and either reporting a negative or an unknown prior HIV test result in the survey. Those not reporting a positive HIV-status were asked about how they perceived their risk for HIV infection and if they knew where they could get tested.

### 2.4. Data Analysis

Data analysis was performed using IBM SPSS Statistics v.24 (IBM, Armonk, NY, USA). Categorical variables were compared using Pearson’s χ^2^ or Fisher’s Exact test when appropriate. Continuous variables were compared using ANOVA test. Bivariable regression analyses were performed to estimate the crude odds ratios (OR) and 95% confidence intervals (CI) of factors associated with each of the three outcome variables: having a reactive test for HIV, having other STI in the past and having TB in the past. Variables significantly associated (*p* < 0.05) were included in the multivariable analyses. Other analyses attempts were conducted with a less conservative *p*-value (<0.2) but no other significant associations were observed. Multivariable logistic regression analyses were performed to estimate the adjusted odds ratios (aOR) and 95% CI of factors associated with a HIV reactive test, with past STI and with past TB.

## 3. Results

### 3.1. Description of the Study Participants

A total of 790 SSAMs were enrolled in the study. The characteristics of the participants are presented in Table 1. Overall, 58.0% were male and the mean age was 38.4 years. Most participants perceived their income as insufficient (77.3%), were non-employed (64.3%) and about 46% had intermediate education (5th–9th grade). Almost 35% were living in the country for 6–15 years.

Of the participants who were sexually active in the last 12 months (*n* = 656, 83.2% of the total sample), the great majority reported inconsistent condom use (82.5%) (Table 1). Almost a fourth reported ≥3 sexual partners and 10.8% engaged in commercial sex (Table 1). Inconsistent condom use was more frequent among women (89.7% vs. 77.7% among men, *p* < 0.001), those reporting only one sexual partner (84.8% vs. 76.6% among those reporting ≥3 partners, *p* < 0.1), and those who did not engage in commercial sex (84.3% vs. 67.6% among those who engaged in commercial sex, *p* < 0.001) (data not shown in table). Inconsistent condom use was independent of age, educational level, occupation, length of stay in Portugal, experience of violence and experience of forced sex.

Overall, 21.9% of the participants reported experience of violence from a partner and 9.6% reported forced sex (Table 1). Reported violence was significantly more prevalent among women (30.4% vs. 15.7% among men, *p* < 0.001), those with insufficient income (24.9% vs. 11.7% among those with sufficient income, *p* < 0.001), with ≥3 sexual partners (27.9% vs. 18.4% among those with one sexual partner, *p* = 0.037), and who engaged in commercial sex (31.0% vs. 19.3% among those who did not engage in commercial sex, *p* = 0.022). Half of participants (49.4%, *n* = 376) did not perceive of themselves as at risk for HIV infection (data not shown in table).

Overall, 49.7% of participants were never tested for HIV (Table 1); of these, almost three quarters (72.6%, *n* = 275) did not know where they could get tested (data not shown in table).

### 3.2. HIV, Other STI and Associated Factors

Forty-three (5.4%) participants had a reactive test for HIV and 132 (16.7%) reported a past STI (Table 1). Of the HIV-positive participants, 8 (18.6%) had other STI in the past.

In the bivariable analysis, prevalence and odds of being HIV positive increased by age (OR 1.06, 95% CI 1.03–1.08; *p* < 0.001), were higher among those with primary/no education (OR 5.52, 95% CI 2.07–14.8; *p* = 0.001), non-employed (OR 2.18, 95% CI 1.03–4.60; *p* = 0.042), with insufficient income (OR 2.98, 95% CI 1.05–8.46; *p* = 0.040), and those who experienced violence from a partner (OR 2.00, 95% CI 1.04–3.83; *p* = 0.037) (Table 1). Inconsistent condom use was less likely to be reported by HIV-positive participants (OR 0.34, 95% CI 0.15–0.76; *p* = 0.009) (Table 1). In the multivariable analysis (Table 2), HIV status remained associated with age, experience of violence from a partner and condom use. 

The bivariable analysis showed that prevalence and odds of having other STI in the past were higher among those with insufficient income (OR 2.82, 95% CI 1.58–5.04; *p* < 0.001) and those living in Portugal for >25 years (OR 2.90, 95% CI 1.56–5.40; *p* = 0.001) (Table 1). Of sexual risk factors, participants who reported ≥3 sexual partners (OR 2.28, 95% CI 1.44–3.61; *p* < 0.001), inconsistent condom use (OR 2.12, 95% CI 1.12–3.99; *p* = 0.021) and engagement in commercial sex (OR 1.93, 95% CI 1.10–3.37; *p* = 0.022) were almost twice as likely to have had past STI, and participants who experienced violence from a partner (OR 2.54, 95% CI 1.70–3.81; *p* < 0.001) and forced sexual relations (OR 3.19, 95% CI 1.91–5.33; *p* < 0.001) were almost three times as likely to have had past STI (Table 1). After adjustment, the odds of having other STI remained higher among participants with insufficient income, living in the country for >25 years, having ≥ 3 sexual partners, reporting inconsistent condom use, having experienced violence from a partner and having had forced sexual relations (Table 2).

### 3.3. TB and Associated Factors

Thirty-two (4.1%) participants reported having had TB in the past. Of the HIV-positive participants, a quarter (25.6%, *n* = 11) reported past TB (Table 1).

In the bivariable analysis, prevalence and odds of past TB increased by age (OR 1.04, 95% CI 1.01–1.06; *p* = 0.002), were higher among non-employed participants (OR 3.11, 95% CI 1.18–8.17; p=0.021), those living in Portugal for >25 years (OR 4.86, 95% CI 1.10–21.4; *p* = 0.037) and being HIV positive (OR 11.9, 95% CI 5.29–26.7; *p* = 0.001), and were lower among women (OR 0.31, 95% CI 0.12–0.75; *p* = 0.010) (Table 1). After adjustment, HIV-positive participants and males remained more likely to have had TB in the past (Table 3).

### 3.4. Unawareness of HIV Status and Testing History

Of the HIV-positive participants, 35% (*n* = 15) were unaware of their HIV infection (8 reported not being infected based on the result of their last HIV test and 7 had never been tested) and the remaining 65% (*n* = 28) reported being infected based on the result of their last HIV test. Most of the participants unaware of their HIV-positive status were female (*n* = 8), reported inconsistent condom use (*n* = 9), had only one sexual partner (*n* = 8) and perceived not being at risk for HIV (*n* = 8). Over a third were living in Portugal for ≤5 years (*n* = 6) and the same proportion were living for >25 years (*n* = 6).

Of the HIV-positive participants unaware of their status, 75% (*n* = 6) had performed their last HIV test over 12 months prior to the survey, while a quarter (*n* = 2/8) had been tested in the previous year, suggesting a recent infection.

Around 37% (*n* = 49) of participants who had other STI were never tested for HIV; of these, 69.4% (*n* = 34) reported not knowing where to get the test. Similarly, 37.5% (*n* = 12) of participants who had TB in their lifetime had never been tested for HIV and 58.3% (*n* = 7) of them did not know where to get tested (data not shown in table).

## 4. Discussion

This study provides valuable data on the burden of infectious diseases and associated factors among sub-Saharan African migrants (SSAMs) in Portugal. Overall, 5.4% of the participants were HIV positive, 16.7% had had other STI in the past and 4.1% had TB in their lifetime. The proportion of HIV found among SSAMs in this study was nine times higher than the estimates for the general Portuguese population (0.6%) [31], and is in line with the HIV prevalence observed among SSAMs in other research in Europe [32]. Of the HIV-positive participants, a quarter (25.6%, *n* = 11) reported past TB, which is not surprising considering that HIV is the main risk factor for TB disease [25]. As shown in previous research, migrant populations are disproportionately affected by HIV-TB co-infection when compared to nationals, with the highest prevalence being observed among migrants originating from African regions [33]. In our study, almost a fifth of HIV-positive participants (18.6%, *n* = 8) also had other STI in the past.

Overall, unprotected sex along with multiple sexual partnerships and engagement in commercial sex were reported by a high proportion of participants. Simultaneous risk behaviors were considerably common among those previously diagnosed with STI. The fact that these participants still engage in risky sexual behaviors, inconsistently use condoms and over a third perceived of themselves as not at risk for HIV, indicate that there may be missed opportunities for prevention of risk behaviors and counselling for behavior change when they are diagnosed with STI.

Even though underreporting cannot be dismissed, the fact that over a third of participants with a reactive HIV test result did not report being HIV-positive suggests a high unawareness of HIV serostatus among SSAMs in this study, surpassing some of the European Union/European Economic Area countries estimations for undiagnosed HIV in the general population [34]. A small group of HIV-positive participants unaware of their status had tested for HIV in the previous year, suggesting recent HIV infection. People who are unaware of their HIV infection are estimated to contribute up to 50–90% of new HIV infections [35]. The proportion of study participants who had ever tested for HIV was notably low (50.3%) and almost a half of the participants unaware of their HIV-positive status had never tested for HIV. These findings are in line with literature documenting high rates of undiagnosed infection and low rates of HIV testing among migrants [36,37], particularly those from sub-Saharan Africa [32,35]. Overall, our findings reflect insufficient testing, potential barriers to testing or testing services not reaching those most at risk [34]. In this community-based participatory research, the community partners’ active involvement in the recruitment of participants, collection of data and provision of HIV rapid test in migrant-friendly community-based settings contributed to a total of 379 out of 790 SSAMs taking the HIV test for the first time; out of 43 participants with reactive test results, 15 became aware of their HIV-status and were referred to health services. This shows that there are alternative approaches, such as community-based participatory outreach initiatives, with potential to reduce barriers to HIV testing and encourage uptake of testing among these populations, which therefore should be promoted [38]. The high proportions of participants with past STI and past TB who never tested for HIV and who did not know where to test indicate that considerable opportunities have been missed for early HIV detection and treatment during contacts with health services, and that integrated care of infectious diseases should be strengthened. Also, as it has been acknowledged, HIV early diagnosis can help prevent onward HIV transmission and acquisition of other STI through the adoption of protective behaviors after notification of HIV infection [39]. In fact, our findings show that HIV-positive participants were more likely to report consistent condom use, reflecting the potential benefits of HIV testing and counselling, as well as follow-up provided at health services in terms of secondary prevention.

Structural factors related to poor living conditions in the host country were found to influence SSAMs vulnerability to infectious diseases. Primary/no educational level, non-employed status and low income characterized a high proportion of the study participants and were significantly associated with HIV, past STI and past TB. These findings are consistent with the extensive literature describing the great influence of social and economic vulnerability in the susceptibility to infectious diseases such as STIs [19,40]. Indeed, socioeconomically vulnerable individuals are positioned in society such that they experience a variety of economic, social, gender, and ethnic-based discriminations that constrain individual agency for sexual decision-making. As a result, these individuals are likely to have more exposure to HIV/STI and a lower capacity to protect against infection [41]. In addition, structurally vulnerable populations often experience co-occurring mutually reinforcing disadvantages, such as limited access to steady employment, and affordable, quality education, which can be barriers in access to health services and trigger limited opportunities to seek healthcare [41,42].

In this study, experience of violence from a partner and forced sex were persistently associated with HIV infection and past STI. The intersection between intimate partner violence and HIV infection has long been documented [43,44]. We found that suffering from intimate partner violence was more prevalent among women, as also evidenced in other research demonstrating that gender norms and relational factors, together with fear of suffering further violence can limit a woman’s ability to refuse sex or negotiate safer sexual behavior [44,45,46,47]. Indeed, a significant group of HIV-positive participants unaware of their status were female, had only one sexual partner and had unprotected sex, indicating that their vulnerability is intrinsically connected to their intimate partners’ risk behavior. Social hardships can aggravate women’s vulnerability by limiting their power and control in the relationship and condom use [46]. Women with reduced socioeconomic resources may depend on partners for economic support, which favors exploitative or abusive situations, sometimes comprising sexual violence, including among female SSAMs [48]. Indeed, participants with lower income reported more frequently having experienced violence from a partner. In addition, a significantly higher proportion of study participants who ever experienced violence had multiple partners and engaged in commercial sex. In fact, studies have shown that abused women are more likely to engage in risky sexual behaviors, increasing their chances of becoming infected with STIs [47,49,50].

The length of stay in Portugal was found to be associated with sexually transmitted infections, with migrants residing for longer time being more likely to report past STI, as found in other studies [51,52]. Of the 43 HIV-positive participants, over a half (*n* = 23) were residing in Portugal for ≥25 years. Migration to the countries of destination usually occur at young age [53]. Therefore, older migrants usually have lived in the host country for a longer time and experienced long-term exposure to risk factors [54]. Consistently, in other studies, acculturation throughout the length of stay was associated with adoption of risky sexual behaviors [55]. But our study also shows that a relevant proportion of the HIV-positive participants (7/43; 16.3%) were residing in Portugal for ≤5 years, and over a third of the HIV-positive participants unaware of their status were similarly recent migrants. These findings show that there are migrant subgroups with disparate levels of vulnerability to infectious diseases associated with multiple interrelated factors. While the burden of infectious diseases among recent migrants may partially be due to the endemicity in their countries of origin, among long-term migrants it may be strongly related to adverse social and living conditions faced during their stay in the host countries [16]. This highlights the complexity of the debate concerning where infectious diseases are acquired, reinforcing that attention must be given to both origin and host countries as contexts of vulnerability to infectious diseases for migrant populations. In Europe, the HIV epidemic among migrants, particularly among those from sub-Saharan African countries, was long assumed to be imported from origin countries with generalized HIV epidemics, which focused prevention efforts mostly on the promotion of HIV testing and early linkage to care [15,32,56]. But as highlighted by other authors, this perception recently changed with increasing evidence of HIV acquisition in Europe [15,32,51]. In the aMASE study, conducted in nine European countries including Portugal, 31% of HIV-infected Africans acquired HIV while living in European host countries, emphasizing the renewed need for primary prevention [57].

In our study, we were also able to identify risk factors associated with TB. Men were more likely to report having had TB in the past, which is consistent with the broad literature on sex differences in TB prevalence [58,59]. In our study, non-employed participants reported more frequently having had TB in the past. Indeed, the poor socio-economic conditions that many migrants face in host countries can increase risk exposure and contribute to differential treatment-seeking behavior, with impact on TB acquisition/reactivation within these communities [60]. In the context of continuing intense migration flows in Europe and given the estimates of high TB incidence in foreign-born populations compared to native populations [58], it has been claimed that migrants and ethnic minorities ‘import’ TB into the European host countries, often being labelled as carriers of infectious agents [60,61]. In contrast, our findings highlight the influence of complex socio-structural factors to increased vulnerability of migrants to TB, reinforcing that European health systems must keep their efforts to guarantee effective social support, besides adequate access to healthcare, in order to strengthen prevention.

Limitations of this study must be mentioned. As the study sample was not randomly recruited, the results may not reflect the situation of SSAMs in general. Notwithstanding, the recruitment strategy used allowed us to gather a large and diverse sample of SSAMs residing in Lisbon. As the data are drawn from a cross-sectional survey, any inferences about causality are not possible. Also, the relatively low numbers of participants reporting infectious diseases may have limited statistical analyses, therefore results must be treated with caution. Participants were asked to report on behaviors over a long duration of time, with possible risk for recall bias. In addition, the use of face-to-face interviews, given the low literacy rate among the study population, may have led to misreporting due to social desirability bias. Nevertheless, in order to minimize potential biases, all interviewers were systematically trained on interview techniques and ethical principles. Self-reported data, in particular on sensitive issues such as HIV status, past STI, past TB and sexual behavior may have potentially resulted in bias. In a worst-case scenario, our data underestimate the burden of infectious diseases and sexual risks among SSAMs.

## 5. Conclusions

This study demonstrates a high burden and vulnerability to infectious diseases among a SSAM population, and high rates of HIV-infection unawareness. Knowledge of HIV status could help protect people from transmitting HIV unknowingly, from suffering unnecessarily from opportunistic infections, and from dying prematurely with no access to treatment [62]. Further efforts are needed to reduce missed opportunities for HIV/STIs prevention and uptake of HIV testing through community-based outreach initiatives and more integrated care of infectious diseases among SSAMs. Social inequalities are persistently a major factor affecting risk for HIV, STIs and TB, and thus addressing social determinants of infectious diseases must remain a public health priority.

## Figures and Tables

**Table 1 ijerph-16-02710-t001:** Bivariable analysis of sociodemographics, sexual behaviors and infections correlated with HIV reactive test, past sexually transmitted infection (STI) and past Tuberculosis infection (TB).

Variable	*n* (%)	HIV Reactive (*n* = 790)	OR (95% CI)	*p*	Past STI (*n* = 790)	OR (95% CI)	*p*	Past TB (*n* = 790)	OR (95% CI)	*p*
		*n* (%)			*n* (%)			*n* (%)		
		43 (5.4)			132 (16.7)			32 (4.1)		
Sociodemographics										
Age (years, mean ± SD) (*n* = 790)	38.4 ± 14.5	50.2 ± 11.2	1.06 (1.03–1.08)	<0.001	40.6 ± 13.6	1.01 (0.99–1.03)	0.063	46.3 ± 13.5	1.04 (1.01–1.06)	0.002
Sex (*n* = 790)										
Male	458 (58.0)	20 (4.4)	1		82 (17.9)	1		26 (5.7)	1	
Female	332 (42.0)	23 (6.9)	1.63 (0.88–3.02)	0.120	50 (15.1)	0.81 (0.55–1.19)	0.291	6 (1.8)	0.31 (0.12–0.75)	0.010
Educational level (*n* = 790)										
Secondary/Higher	219 (27.7)	5 (2.3)	1		39 (17.8)	1		10 (4.6)	1	
Intermediate	361 (45.7)	14 (3.9)	1.73 (0.61–4.86)	0.301	58 (16.1)	0.88 (0.57–1.38)	0.586	9 (2.5)	0.72 (0.31–1.69)	0.457
Primary/no education	210 (26.6)	24 (11.4)	5.52 (2.07–14.8)	0.001	35 (16.7)	0.92 (0.56–1.52)	0.754	13 (6.2)	1.87 (0.75–4.68)	0.180
Occupation (*n* = 790)										
Employed	282 (35.7)	9 (3.2)	1		39 (13.8)	1		5 (1.8)	1	
Non-employed	508 (64.3)	34 (6.7)	2.18 (1.03–4.60)	0.042	93 (18.3)	1.40 (0.93–2.10)	0.107	27 (5.3)	3.11 (1.18–8.17)	0.021
Perceived income (*n* = 790)										
Sufficient	179 (22.7)	4 (2.2)	1		14 (7.8)	1		5 (2.8)	1	
Insufficient	611 (77.3)	39 (6.4)	2.98 (1.05–8.46)	0.040	118 (19.3)	2.82 (1.58–5.04)	<0.001	27 (4.4)	1.61 (0.61–4.24)	0.336
Length of stay in the country (*n* = 790)										
≤5 years	155 (19.6)	7 (4.5)	1		14 (9.0)	1		2 (1.3)	1	
6–15 years	276 (34.9)	10 (3.6)	0.80 (0.30–2.13)	0.648	49 (17.8)	2.17 (1.16–4.08)	0.016	11 (4.0)	3.18 (0.70–14.5)	0.136
16–25 years	91 (11.5)	3 (3.3)	0.72 (0.18–2.86)	0.641	9 (9.9)	1.10 (0.46–2.67)	0.823	3 (3.3)	2.61 (0.43–15.9)	0.299
>25 years	268 (33.9)	23 (8.6)	1.99 (0.83–4.74)	0.123	60 (22.4)	2.90 (1.56–5.40)	0.001	16 (6.0)	4.86 (1.10–21.4)	0.037
Sexual practices										
Number of sexual partners (last 12 months; *n* = 655)										
1	376 (57.4)	18 (4.8)	1		53 (14.1)	1				
2	125 (19.1)	3 (2.4)	0.48 (0.14–1.69)	0.258	24 (19.2)	1.45 (0.85–2.46)	0.172	-	-	-
≥3	154 (23.5)	6 (3.9)	0.81 (0.31–2.07)	0.655	42 (27.3)	2.28 (1.44–3.61)	<0.001	-	-	-
Condom use (last 12 months, *n* = 656)										
Consistent	115 (17.5)	10 (8.7)	1		12 (10.4)	1		-	-	-
Inconsistent	541 (82.5)	17 (3.1)	0.34 (0.15–0.76)	0.009	107 (19.8)	2.12 (1.12–3.99)	0.021	-	-	-
Engaged into commercial sex (last 12 months; *n* = 656)										
No	585 (89.2)	22 (3.8)	1		99 (16.9)	1		-	-	-
Yes	71 (10.8)	5 (7.0)	1.94 (0.71–5.29)	0.196	20 (28.2)	1.93 (1.10–3.37)	0.022	-	-	-
Experience of violence from partner ever (*n* = 790)										
No	617 (78.1)	28 (4.5)	1		83 (13.5)	1		-	-	-
Yes	173 (21.9)	15 (8.7)	2.00 (1.04–3.83)	0.037	49 (28.3)	2.54 (1.70–3.81)	<0.001	-	-	-
Experience of forced sexual relations ever (*n* = 789)										
No	713 (90.4)	37 (5.2)	1		105 (14.7)	1		-	-	-
Yes	76 (9.6)	6 (7.9)	1.57 (0.64–3.84)	0.327	27 (35.5)	3.19 (1.91–5.33)	<0.001	-	-	-
HIV testing										
HIV reactive (*n* = 790)										
No	747 (94.6)	-	-	-	124 (16.6)	1	0.732	21 (2.8)	1	
Yes	43 (5.4)	-	-	-	8 (18.6)	1.15 (0.52–2.54)		11 (25.6)	11.9 (5.29–26.7)	0.001
Ever tested for HIV (*n* = 763)										
Yes	384 (50.3)	36 (9.4)	-		80 (20.8)	1		17 (4.4)	1	
No	379 (49.7)	6 (1.6)	-	-	49 (12.9)	0.56 (0.38–0.83)	0.004	12 (3.2)	0.71 (0.33–1.50)	0.365

**Table 2 ijerph-16-02710-t002:** Factors associated with HIV reactive test and past STI.

Variable	HIV Reactive		Past STI	
	Adjusted OR (95% CI)	*p*	Adjusted OR (95% CI)	*p*
Age (years)	1.07 (1.03–1.11)	<0.001	1.01 (0.99–1.03)	0.310
Sex				
Male	1		1	
Female	1.83 (0.70–4.74)	0.215	0.96 (0.57–1.61)	0.865
Education				
Secondary/Higher	1			
Intermediate	1.12 (0.33–3.83)	0.860	-	-
Primary/no education	2.19 (0.58–8.37)	0.250	-	
Occupation				
Employed	1		-	-
Non-employed	1.98 (0.67–5.86)	0.220	-	
Perceived income				
Sufficient	1		1	
Insufficient	0.62 (0.19–2.07)	0.439	2.86 (1.48–5.56)	0.002
Length of stay in the country				
≤5 years	-		1	
6–15 years	-	-	1.54 (0.77–3.10)	0.221
16–25 years	-	-	0.99 (0.38–2.61)	0.987
>25 years	-	-	2.16 (1.06–4.38)	0.033
Number of sexual partners (last 12 months)				
1	-		1	
2	-	-	1.50 (0.85–2.65)	0.165
≥3	-	-	2.06 (1.14–3.70)	0.016
Engaged into commercial sex (last 12 months)				
No	-		1	
Yes	-	-	0.98 (0.49–1.98)	0.965
Condom use (last 12 months)				
Consistent	1		1	
Inconsistent	0.23 (0.09–0.58)	0.002	2.61 (1.32–5.17)	0.006
Physical and/or emotional abuse from partner				
No	1		1	
Yes	2.77 (1.08–7.10)	0.034	1.80 (1.09–3.00)	0.022
Forced sexual relations				
No	-		1	
Yes	-	-	2.57 (1.33–4.96)	0.005

**Table 3 ijerph-16-02710-t003:** Factors associated with history of TB disease.

Variable	Past TB	
	Adjusted OR (95% CI)	*p*
Age (years)	1.01 (0.99–1.04)	0.349
Sex		
Female	1	
Male	3.42 (1.29–9.06)	0.014
Occupation		
Employed	1	
Non-employed	1.92 (0.68–5.38)	0.217
Length of stay in the country		
≤5 years	1	
6–15 years	2.96 (0.62–14.16)	0.173
16–25 years	2.56 (0.40–16.46)	0.322
>25 years	2.58 (0.54–12.32)	0.235
Reactive test for HIV		
No	1	
Yes	11.48 (4.55–28.94)	< 0.001

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
