# Peer review of "Are Opportunities Being Missed? Burden of HIV, STI and TB, and Unawareness of HIV among African Migrants"

_ijerph, 2019, doi:10.3390/ijerph16152710_

Round 1

Reviewer 1 Report

Thank you for giving me the opportunity to review your submission. The data seem sound and possibly reflect a need for more targeted prevention services for this underserved community. However, the sampling method significantly affects the generalizability of these results. This has been acknowledged as a weakness but remains  a significant limitation. To this end, given the reportedly large number of SSAM's in Lisbon, it would be helpful to know what proportion were included in the study ie how many SSAM's are living in Lisbon/ Portugal. Overall, the manuscript is clear but would benefit from another review to correct repetition and ambiguous  language. Lastly, it seems counter-intuitive that inconsistent condom use was relatively protective for HIV infection. Any ideas why this may be so?

Reviewer 2 Report

Review of IJERPH 554231

Thank you for the opportunity to review “Are there opportunities being missed? Burden of HIV, STI and TB, unawareness of HIV and testing among SSAMs” in which the authors review predictors of HIV, STI and TB testing among African migrants to the EU, Portugal in particular.  This is an interesting study and I believe with some minor edits could make an interesting submission.  In general, the English was excellent; however, I would recommend another review of the paper to check for minor grammatical issues.

My specific comments follow:   

Your title is a little awkward and includes a confusing acronym.  I think you could convey the same information in something a little shorter, like “Are there opportunities being missed? Burden of HIV, STI and TB among African migrants to the EU [or Portugal?]”

Line 84: “A total of 790 SSAMs were enrolled in the study” This is results, not methods

Line 101: “n=656” this is also results, not methods

Line 120: was there a reason you set your p value so restrictive (p<0.05) when considering which variables to include in the multivariable analysis?  This is typically a bit more relaxed (p<0.2 or <0.15) in the event a borderline predictor becomes “significant” in the context of other variables.  Also, given the small number of outcomes in some cases (eg, only about 30 cases of TB), I would recommend reconsidering this

Your outcome is a combination of HIV, past STI or past TB (any, right?).  While HIV and STI makes sense – these are all sexually transmitted, TB is very different.  What is the rationale for including TB as part of your outcome?  I’m not sure this makes sense, and likely affects your model’s validity.  Edit: now I see it looks like you did three analyses, one for each outcome.  Is this correct?  Please clarify this in your methods (and you could consider collapsing HIV+STI, maybe, but keep TB separate).  With only 43 HIV cases, and 32 reporting TB, you have very little power to do any sort of robust analyses on these outcomes.  This low statistical power should be noted as a limitation

Results: how many were screened and did not participate?  Why did they not participate?  This also is a limitation and a high rate of screen-outs or refusals affects your generalizability to other SSAMs in Portugal or the EU.  Consider including a CONSORT style flow diagram (http://www.consort-statement.org/consort-statement/flow-diagram) except without the randomization.  See also STROBE statement (https://www.strobe-statement.org)
